## Research Article

mental health literacy; mental health promotion; adaptation; the Kenyan guide; sub-Saharan Africa

**Corresponding author:**
Patrick N. Mwangala;
Email: patrick.nzivo@aku.edu

# Feasibility, acceptability and initial efficacy of a community-based mental health literacy program delivered by civil society organizations among adults in Kenya: A quasi-experimental study

Patrick N. Mwangala[1] , Venoranda Rebecca Kuboka[2], Nimo Sharif[1], Ann Karendi[1], Gideon Mbithi[1] and Amina Abubakar[1]

[1]Institute for Human Development, The Aga Khan University, Nairobi, Kenya and [2]Youth Changers Kenya, Kenya

## Abstract

Mental health literacy (MHL) strategies are crucial for mental health promotion and prevention. This study aimed to determine the acceptability, feasibility, appropriateness and initial efficacy of an adapted MHL program in a community sample of adults in Kenya. This was a quasi-experimental pre-post study conducted from July 2023 through July 2024. The MHL program contained nine modules delivered over 3 days. Participants were assessed at baseline and immediately after the program. The primary outcomes were mental health knowledge and participants' attitudes on mental health/illness. Secondary outcomes included depressive symptoms, anxiety symptoms, self-perceived social support, self-perceived wellbeing, MHL program acceptability, feasibility and appropriateness. Relative to baseline, we observed statistically significant improvement in mental health knowledge and attitude on mental health/illness postintervention. We also observed significant improvements in all secondary outcomes. The MHL program also emerged as contextually appropriate, acceptable and feasible. The adapted MHL program is acceptable and appropriate and can feasibly be delivered by trained non-specialist facilitators. Also, the MHL program has the potential to increase participants' MHL and attitudes and reduce symptoms of common mental disorders and promote self-perceived wellbeing. Future research should explore how improvements can be sustained over the long term.

## Impact statements

Mental health literacy (MHL) interventions play a crucial role in enhancing individuals' ability to recognize, understand and manage their mental health problems, reducing stigma, promoting early help-seeking and empowering them to maintain their well-being through self-help and informed professional support. Unfortunately, many communities in developing countries have inadequate MHL. Poor MHL has been associated with negative health outcomes, including reduced use of preventive services, minimal adherence to prescribed treatment, increased risk of hospitalization which results in higher healthcare costs and increased risks for other illnesses. The high burden of mental disorders, the treatment gap, and the scarcity of mental health resources in sub-Saharan Africa (SSA) highlight the critical need for optimizing MHL among individuals in the region. To bridge this gap in Kenya, the current study evaluates the acceptability, appropriateness, feasibility and initial efficacy of an adapted MHL program delivered by trained non-specialist facilitators drawn from local grassroots organizations. This approach holds promise in scaling up MHL work in low-resource settings e.g. through integration into grassroots organizations, task-shifting and utilizing local solutions/expertise. This work sets the foundation for scaling community MHL programs in Kenya and offers a model for similar adaptations to other low-resource settings.

## Introduction

Mental health literacy (MHL) was first described by Jorm and colleagues (Jorm et al. 1997, p.182) as the 'knowledge and beliefs about mental disorders which aid their recognition, management, or prevention'. Over the years, this MHL concept has evolved to incorporate four components: (a) understanding how to obtain and maintain positive mental health, (b) understanding mental health illnesses and their treatments, (c) reducing mental health related stigma and (d) promoting help-seeking (Kutcher et al., 2016). Research evidence to-date confirms that MHL is associated with enhanced identification of mental illnesses and improved attitudes and intended behaviors

toward people with mental illnesses (Milin et al., 2016) as well as improvements in social skills and attitudes toward help giving (Lo et al., 2018) and help seeking (Mumbauer-Pisano and Barden, 2020). Conversely, poor health literacy has been associated with adverse outcomes including minimal use of preventive services (Santos et al., 2017), reduced adherence to prescribed treatment (Miller, 2016), increased risk of hospitalization (Baker et al., 1998), causing higher healthcare costs (Greene et al., 2019) and elevated risk of other illnesses (Mantell et al., 2020). This underscores the crucial role of understanding and improving the MHL of different populations, especially those residing in low- and middle-income countries (LMICs) who carry the greatest burden of mental health problems.

Previous research on MHL has primarily focused on children and adolescents (Frețian et al., 2021), students (Amado-Rodriguez et al., 2022; Reis et al., 2022; Liao et al., 2023), teachers, educators (Johnson et al., 2023) and various community members in high-income countries (HICs) (Andrade et al., 2022). While this body of evidence has significantly contributed to our understanding of MHL, the underrepresentation of LMICs is concerning for a few reasons: (i) three-quarters of the global mental health burden comes from LMICs, and predictive models suggest an exponential rise in this burden in certain regions e.g. SSA (Charlson et al., 2014); (ii) scarce mental health infrastructure and resources; (iii) the quality and availability of professional mental health services are poor; (iv) several countries have non-existent, inappropriate or deficient mental healthcare policies and lack national strategies to promote/prevent mental health problems and (v) big mental health treatment gap (>75%). To the best of our knowledge, only two reviews have attempted to summarize the state of MHL in SSA. The first review (Atilola, 2015), included 19 studies across eight countries. The authors concluded that the MHL studies in SSA were few, too narrow in scope, too limited in their distribution and too restricted in their perspectives for comprehensive understanding of this concept. The other review (Letsoalo et al., 2025) included 14 studies that were published between 2015 and 2023. Similarly, many of the included studies reported low levels of MHL. Closer home in Kenya, the few MHL studies in the country have been conducted among primary healthcare workers (Marangu et al., 2021; Goyal et al., 2023; Muriuki et al., 2024) and adolescents (Ayiro et al., 2023; Wadende and Sodi, 2023). Overall, the high burden of mental health problems, scarcity of mental health resources and low MHL underscores the need for programs/interventions that improve MHL in different populations.

In recent years, there has been a renewed commitment to develop and implement programs to promote MHL in the region. Such programs are varied and include family-orientated, school-orientated, peer-orientated, online orientated (Bruand et al., 2024; Sodi et al., 2025) and community based interventions (Mutiso et al., 2019; Arthur et al., 2022). Most of the available MHL interventions in SSA have targeted young people in schools (Arthur et al., 2022). The other populations studied include community leaders (Arthur et al., 2022), parents (Puffer et al., 2016; Alemu et al., 2023), teachers (Kutcher et al., 2016), healthcare providers (Bruand et al., 2024) and pregnant women (Tessema et al., 2025). The findings from these studies suggest that MHL interventions can increase mental health knowledge, promote non-stigmatizing attitudes and improve attitudes toward help-seeking behavior among the different studied populations. Nonetheless, the results are not consistent across all the different components of MHL, and training itself is not standardized, with a great variation in practices and implementation strategies.

Intervention studies to improve MHL, particularly among adult Kenyans residing in urban and rural informal settlements, are lacking despite their heightened vulnerability to common mental health problems. Thus, the primary objective of this study is to examine whether an adapted MHL intervention (Kutcher et al., 2013) (*the Guide* – Kenyan version) generates improvements in mental health knowledge and attitudes toward mental health and illness among adults in urban and rural informal settlements in Kenya. The secondary goals are to examine the impact of the MHL intervention on participants' psychosocial functioning (depressive and anxiety symptoms, social support and general wellbeing) post-intervention; as well as evaluating the appropriateness, acceptability and feasibility of the adapted MHL intervention.

## Methods

### Study design and setting

This was a quasi-experimental pre–post study conducted between July 2023 and July 2024 across three Kenyan counties: Nairobi, Mombasa and Kwale. The data utilized in this study is part of a larger mental health project, *Advancing Gender Equality through Civil Society (AGECS)*, being implemented at the Aga Khan University – Institute for Human Development (AKU-IHD) in Kenya. The primary aim of the AGECS Mental Health Project was to understand the burden and underlying factors of mental health problems among women and their spouses, and subsequently design, implement and assess the impact of mental health interventions across the three counties. Further details of the project have been described elsewhere (Mwangala et al., 2024; Mwangala et al., 2025). Nairobi, Mombasa and Kwale counties are part of the 47 devolved units of government in Kenya which came into effect in 2013. Mombasa and Nairobi counties are largely urban settings while Kwale is predominantly a rural setting. In Nairobi, the study was carried out in Westlands subcounty, while in Mombasa, the study was conducted in Changamwe subcounty. In Kwale, the study was conducted in Matuga subcounty.

### Study participants and sampling

Our target population in this study was adult men and women aged at least 18 years and resident in the three counties of interest. Potential respondents were recruited using sequential sampling from households and public places e.g. markets and workplaces. Recruitment was conducted by representatives of local civil society organizations (CSOs) – subawarded to implement the project in the community. To be included, participants had to ≥18 years, able to provide written informed consent and able to speak Swahili or English. Children and adolescents were excluded because we had a different project dedicated to examining the same issues in this age cohort. The current study was part of a larger project that aimed to advance gender equality and address challenges faced by women (Mwangala et al., 2025). As such, we intentionally sampled more women.

### Overview of the MHL intervention and the adaptation process

The *Kenyan Guide* community-based MHL program is an adaptation and modification of the original Mental Health and High School Curriculum Guide *(The Guide)*. The original *Guide* is a school-based MHL resource developed by mental health experts, educators and the Psychiatric Unit of Health in Canada (Kutcher

et al., 2013). It consists of six modules including: (a) stigma of mental illness, (b) understanding mental health and mental illness, (c) information on specific mental illnesses, (d) experiences of mental illness, (e) seeking help and finding support and (f) importance of positive mental health. The *Guide* has been adapted for use in SSA including among Malawian university students and young adults (Nyali et al., 2025), students and educators in Tanzania (Kutcher et al., 2017), students in Ethiopia (Hassen et al., 2022) and school-going adolescents in Kenya.

In the current project, we sought to adapt the *Guide* for community-based implementation in Kenya, targeting adults in urban and rural informal settlements in Kenya. Several elements were considered when adapting the *Guide* to the Kenyan context including language, content, local conceptualizations of concepts, and the delivery process. Prior permission was sought from the developers to adapt the *Guide* in Kenya. We then held two separate adaptation (*co-design*) meetings which lasted 6 days in total. The two co-design sessions brought together different stakeholders including: a global mental health researcher, psychologists, clinicians, ministry of health officials, representatives of CSOs, psychiatric nurses, community health liaison persons and religious leaders. The participants critically evaluated the *Guide* in terms of content (e.g. strategies and accompanying activities), and delivery method (e.g. sessions, duration, delivery agents) and deliberated the potential barriers and facilitators during the community implementation of the adapted *Kenyan Guide*. Supplementary File 1 provides a comparison between the original *Guide* and the adapted *Kenyan Guide*. The final adapted *Kenyan Guide* consisted of nine modules as highlighted in Figure 1. The choice and design of this MHL was partly informed by the formative phase results of the parent AGECS project which have been published elsewhere (Mwangala et al., 2025).

### Intervention delivery agents

The adapted *Kenyan Guide* was delivered by trained CSO representatives/facilitators who were staff or volunteers of the respective 7 CSOs that were subawarded to implement the MHL program in their respective communities. The grassroots organizations were identified through a mapping exercise conducted across the three counties of interest. The mapping exercise followed the steps recommended by Price et al. (2019). This exercise yielded 25 CSOs across the three counties. Each of the 25 CSOs provided 3 representatives for the 2-week training on the adapted *Kenyan Guide* (5 days classroom training and 5 days in-field practice). The training program was delivered by a mental health expert. An additional three mental health researchers (one global mental health practitioner and two psychologists), who would provide supervision to the CSO representatives, were trained. The training included concepts of mental health, state of mental health in Kenya, overview of the parent project, principles of research ethics, tenets of the adapted *Kenyan Guide*, the adaptations made and its proposed delivery, psychological first aid, basic training of trainers, pre-and post-survey training and CSOs presentations on their respective organizations. Following the classroom training, the CSOs representatives took part in a 5-day field practice in the respective communities. Subsequently, seven CSOs were selected for sub-award and community implementation of the adapted *Kenyan Guide* (three in Nairobi, two in Kwale and two in Mombasa). The sub-award was based on the CSOs performance throughout the training and pilot period, funds availability, in addition to a due diligence exercise at the organization's office.

### Competence and intervention fidelity

During the classroom training of the MHL program, competency was reinforced through several methods, including group activities, discussions, individual reading assignments, and role play. The facilitators also received supervision and on the spot feedback when implementing the *Kenyan Guide* to ensure adherence with the program structure and components. The supervisors (project officers with a background in psychology) observed all the sessions (at least the first and last day) of the MHL program and provided

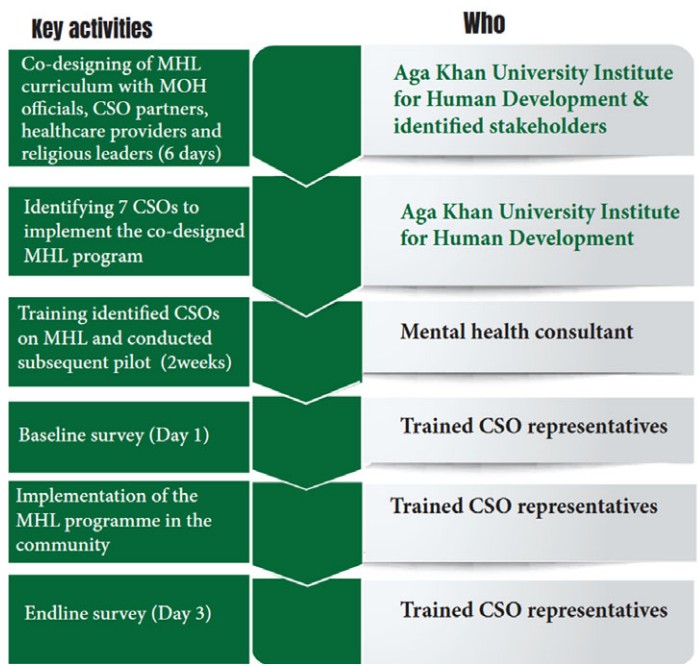
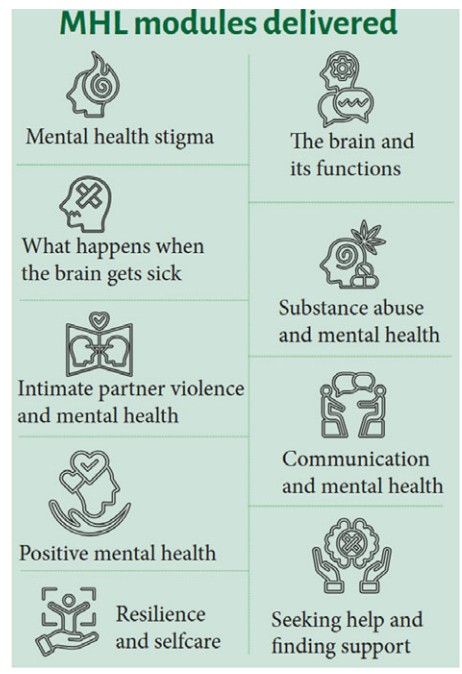

**Figure 1.** Kenyan Guide adaptation and implementation process.

feedback at the end using a structured adherence checklist. For additional support, progress monitoring and brainstorming of additional concerns – the project officers held a virtual meeting to resolve the arising issues.

### Data collection procedure

The MHL program was implemented in various community spaces including churches, public schools, social halls, mosques and health centers (in some instances, a small fee was paid to the management to access and utilize the facilities). Across the three counties, the MHL program was implemented in groups that depended on the size of the community venue (typically, a group comprised of 26 to 53 participants). Each class was facilitated by at least two trained CSO representatives. Additionally, at least one project officer from AKU-IHD was present during the MHL program implementation to monitor the implementation fidelity, as well as offer support as necessary. The program was implemented in 3 days (usually from around 8:00 am in the morning to about 1:00 pm in the afternoon). In between sessions, participants were given short breaks. On the first day of the MHL program, the participants were taken through the informed consenting process and did the baseline assessments before the program started. Endline assessments were completed on day 3 of the MHL program, immediately after the last module. All baseline and endline assessments were paper and pencil. Participants were provided with water and snacks each day and reimbursed their out-of-pocket expenses including the fare.

### Measures

#### Sociodemographic information

Participants were asked to provide demographic information at baseline, including age, sex, highest level of education attained, religious affiliation, marital status and current employment status. Respondent's socioeconomic status (SES) was evaluated using an asset index that has previously been utilized in Kenya (Abubakar et al., 2008). The tool screens for the ownership of disposable assets such as television and motorbike. A single score is generated by adding the list of assets, with a higher score translating to a higher SES.

#### Primary outcome measures

**Primary outcome measures** were administered to participants at baseline (day 1) and endline (day 3) and included the Knowledge and Attitude on Mental Health questionnaires. These questionnaires were adapted from the Student Survey questionnaire developed by the Canadian creators of *the Guide* (Wei et al., 2023). The Mental Health Knowledge Questionnaire includes 20 items addressing basic understanding about the brain, common mental health problems, causes of mental illness and treatment. Participants were required to choose one of three options: 'True', 'False' or 'Do not know'. Each correct response received one point for a total possible score of 20. The questionnaire demonstrated good internal consistency in this study with a Cronbach's α of 0.78. The other primary measure, Attitude to Mental Health Questionnaire, comprised of 25 items and required participants to respond to each item on a 7-point Likert scale, indicating how strongly they agree or disagree: 'strongly disagree', 'disagree', 'disagree a little', 'not sure', 'agree a little', 'agree' and 'strongly agree'. For items 1, 2, 3, 8, 10, 11, 13, 14, 15, 16, 20, 21, 22, 23, 24 and 25 – 'strongly disagree' response

was scored 3 points, 'disagree' was scored 2 points, 'disagree a little' was given 1 point, and the rest of the Likert categories awarded 0 points. The remaining items were reverse coded. Total scores ranged from 0 to 75. Likewise, the attitude to mental health questionnaire yielded good internal consistency with a Cronbach's α of 0.75.

#### Secondary outcome measures

**Measures of secondary outcomes** (*symptoms of depression and anxiety, perceived wellbeing, social support and the acceptability, appropriateness and feasibility of MHL intervention*) included the (a) 9-item Patient Health Questionnaire (PHQ-9) (Kroenke et al., 2001), (b) 7-item Generalized Anxiety Disorder Scale (GAD-7) (Spitzer et al., 2006), (c) World Health Organization-Five Item Well-Being Index (WHO-5) (Topp et al., 2015), (d) Multidimensional Scale of Perceived Social Support (MSPSS) (Zimet et al., 1988), (e) Acceptability of Intervention Measure (AIM) (Weiner et al., 2017), (f) Intervention Appropriateness Measure (IAM) (Weiner et al., 2017) and (g) Feasibility Intervention Measure (FIM) (Weiner et al., 2017), respectively. Additionally, we administered a 9-item exit checklist documenting the participants' experiences with the MHL intervention. PHQ-9 (Mwangi et al., 2020), GAD-7 (Nyongesa et al., 2020) and WHO-5 (Chongwo et al., 2018) have been locally adapted and validated in Kenya. The rest of the secondary measures were adapted before being utilized in the current project, yielding good internal consistency, Cronbach α values of 0.85–0.90. The adaptation process of measures followed international guidelines for translation of tools in health research which includes these steps: forward translation by bilinguals, an expert panel review, back-translation and pre-testing (pilot testing/cognitive interviews) with the target population to ensure cultural relevance, clarity and accuracy (Cruchinho et al., 2024). For this study, measures were independently translated from English to Swahili by two staff members who were fluent in both languages. After that, back-translation into English was done by another independent pair of translators. A panel of Kenyan researchers, knowledgeable about the culture and fluent in both English and Swahili and the translators held a harmonization meeting to ensure conceptual, content, semantic and idiomatic equivalence of the tools. In all instances, total scores of these measures were analyzed as continuous outcomes. PHQ-9, GAD-7, WHO-5 and MSPSS were administered at baseline (Day 1) and endline (Day 3).

#### Sample size

This study was primarily formative, with the intention of assessing the acceptability, appropriateness and potential efficacy of the adapted MHL program as delivered by trained non-specialist facilitators from the local communities in this population. Thus, no formal sample size and power calculations were done. The chosen sample size choice was informed by previous exploration work in similar settings in Africa (Kutcher et al., 2016; Kutcher et al., 2017). Feasibility assessment of key components such as training, delivery and supervision requires non-specialist facilitators to have sufficient caseload. The chosen sample size allowed each grassroot organization to have a minimum of two classes of participants (totaling about 100 clients), thus offering a realistic scenario for assessing the intervention's implementation while remaining manageable for non-specialist facilitators who require ongoing support and supervision.

## Data analysis

Data analysis was carried out in STATA version 17 for Windows (StataCorp LP, College Station, Texas, USA). Descriptive statistics e.g. frequencies, percentages, and means (standard deviations) were utilized to summarize the sample characteristics, as well as the acceptability, appropriateness and feasibility of the MHL program. We used paired sample t-tests to evaluate the effect of the MHL program. Cohen's d was calculated to determine the effect sizes of the intervention effect (Lakens, 2013). We also adopted the random intercept model to account for correlation between the repeated measurements (that is mental health knowledge scores, mental health attitude scores, depressive symptoms, anxiety symptoms, perceived social support and perceived wellbeing scores) by including the individual study identifiers as a random intercept. The random intercept model was used to assess the association of pre-intervention sociodemographic characteristics with the respective primary and secondary outcomes.

## Results

### Participants characteristics

Table 1 summarizes the sociodemographic characteristics of the study participants at baseline. A total of 638 participants (64.3% females) took part in the study across the three counties. The mean age of the respondents was 34 years (sd = 13.1). Majority of the participants were unemployed (60%), Christians (62%) and had basic education (65%). Women were more likely to have lower educational attainment, asset index (surrogate measure of socioeconomic status) and more likely to be unemployed compared to their male counterparts.

### Primary outcomes

#### Knowledge and attitude on mental health

The mean change in Mental Health Knowledge scores from baseline to endline was −1.68; 95% CI −1.92 to −1.43. This difference was statistically significant; $p < 0.001$. The effect size, as measured by Cohen's *d*, was 0.45, indicating a medium effect (see Table 2). A larger and greater improvement was observed for Attitude scores where the mean change from baseline to endline was −5.02; 95% CI −6.09 to −3.96. The difference was also statistically significant; $p = <0.001$; Cohen's *d* = 1.96, translating into a large effect.

### Secondary outcomes

Similar patterns of improvement were observed for all secondary outcomes (Table 2). The mean change (SD) in PHQ-9 score from

**Table 1.** Baseline characteristics of the study population by gender, *n* = 638

| Characteristic | Total sample *n = 638* | Gender | | |
| --- | --- | --- | --- | --- |
| | | Females, n = 410 | Males, n = 228 | *p*-value |
| **Sociodemographic** | | | | |
| **Study site** | | | | |
| *Nairobi* | 246 (38.6%) | 168 (41.0%) | 78 (34.2%) | **0.009** |
| *Mombasa* | 198 (31.0%) | 110 (26.8%) | 88 (38.6%) | |
| *Kwale* | 194 (30.4%) | 132 (32.2%) | 62 (27.2%) | |
| **Age – Mean (SD)**, *missing = 33* | 34.3 (13.1) | 34.7 (13.2) | 33.6 (13.0) | 0.33 |
| **Level of education**, *missing = 2* | | | | |
| *Tertiary* | 199 (31.3%) | 116 (28.4%) | 83 (36.6%) | **<0.001[β]** |
| *Secondary* | 278 (43.7%) | 166 (40.6%) | 112 (49.3%) | |
| *Primary* | 138 (21.7%) | 107 (26.1%) | 31 (13.7%) | |
| *None* | 21 (3.3%) | 20 (4.9%) | 1 (0.4%) | |
| **Employment**, *missing = 1* | | | | |
| *Skilled/Professional* | 97 (15.2%) | 42 (10.3%) | 55 (24.1%) | **<0.001** |
| *Unskilled/casual* | 158 (24.8%) | 104 (25.4%) | 54 (23.7%) | |
| *Unemployed* | 382 (60.0%) | 263 (64.3%) | 119 (52.2%) | |
| **Marital status**, *missing = 1* | | | | |
| *Never married* | 280 (44.0%) | 166 (40.6%) | 114 (50.0%) | **<0.001** |
| *Separated/Divorced/Widowed* | 107 (16.8%) | 86 (21.0%) | 21 (9.2%) | |
| *Married/cohabiting* | 250 (39.2%) | 157 (38.4%) | 93 (40.8%) | |
| **Religion**, *missing = 1* | | | | |
| *Christian* | 395 (62.0%) | 250 (61.1%) | 145 (63.6%) | 0.54 |
| *Muslim* | 242 (38.0%) | 159 (38.9%) | 83 (36.4%) | |
| **Asset index score – mean (SD)** | 3.9 (2.1) | 3.6 (2.0) | 4.4 (2.1) | **<0.001** |

β Based on Fisher's exact test. **SD**, standard deviation. Bold p-values are those which were significant at the ≤0.05 level.

**Table 2.** Mean primary and secondary outcomes at baseline and post-intervention across the study

| Outcome | Baseline score M(SD) | Endline score M(SD) | Cohen's d | p-value |
|---|---|---|---|---|
| **Primary outcomes** | | | | |
| Knowledge of Mental Health Scores | 11.1 (3.0) | 12.8 (2.3) | 0.45 | **<0.001** |
| Attitudes to Mental Health scores | 32.8 (12.1) | 37.8 (13.7) | 1.96 | **<0.001** |
| **Secondary outcomes** | | | | |
| Depressive symptoms (PHQ9) | 8.3 (5.6) | 7.2 (5.1) | 0.71 | **<0.001** |
| Anxiety symptoms (GAD7) | 6.9 (4.7) | 6.2 (4.5) | 0.60 | **<0.001** |
| Perceived social support (MSPSS) | 58.4 (16.1) | 61.9 (13.9) | 2.03 | **<0.001** |
| Perceived wellbeing (WHO–5 Well-Being Index) | 13.6 (6.3) | 14.9 (6.3) | 0.92 | **<0.001** |

For Knowledge and Attitude on Mental Health, MSPSS and WHO-5 Wellbeing Index scores, higher scores indicate better/improved outcomes; while for GAD7 and PHQ9, lower scores indicate better mental wellbeing. PHQ9, Patient Health Questionnaire (nine items); GAD7, Generalized Anxiety Disorder Scale (seven items); MSPSS, Multidimensional Scale of Perceived Social Support (12 items); WHO-5 Well-Being Index, World Health Organization Wellbeing Index (five items); SD, Standard deviation. Bold p-values are those which were significant at the ≤0.05 level.

baseline to endline was 1.0 (5.0) while that of GAD-7 score was 0.68 (4.2). On the other hand, the mean change (SD) in WHO-5 Wellbeing Index score from baseline to endline was – 1.3 (6.5) while that of perceived social support was −3.5 (14.3). All the differences in scores were statistically significant ($p < 0.001$) and exhibited medium to large effect sizes (Table 2).

### Associations of primary and secondary outcomes with baseline assessment predictors

Table 3 summarizes estimates and corresponding 95% confidence intervals (CIs) obtained from the random intercept model. The random intercept model for mental health knowledge ($r = 0.22$), mental health attitude ($r = 0.42$), depressive symptoms ($r = 0.55$), anxiety symptoms ($r = 0.57$), wellbeing index ($r = 0.42$) and social support ($r = 0.53$) showed observations within participants were positively correlated. Overall, we observed statistically significant improvements in all primary and secondary outcomes post-intervention relative to pre-intervention: mental health knowledge ($\beta = 1.68$, 95% CI [1.45–1.92], $p < 0.001$), mental health attitude ($\beta = 5.18$, 95% CI [4.12–6.23], $p < 0.001$), depressive symptoms ($\beta = -1.22$, 95% CI [−1.60 to −0.84], $p < 0.001$), anxiety symptoms ($\beta = -0.71$, 95% CI [−1.04 to −0.38], $p < 0.001$), social support ($\beta = 3.33$, 95% CI [2.25–4.41], $p < 0.001$) and wellbeing index ($\beta = 1.32$, 95% CI [0.81–1.83], $p < 0.001$). Urban study sites (relative to rural), being male, secondary and tertiary levels of education (relative to none) and higher socioeconomic status were significantly positively associated with mental health knowledge scores. On the other hand, older age, tertiary educational level (relative to none) and urban study sites (relative to rural) were significantly positively associated with mental health attitude scores. Among secondary outcomes, study site, participant sex and socioeconomic status were found to be significant predictors for depressive and anxiety symptoms. Significant predictors for

perceived wellbeing included study site, participant sex, level of education and socioeconomic status. Significant predictors for perceived social support included study site and participant socioeconomic status.

### Acceptability, appropriateness and feasibility of the MHL program and user satisfaction

Measures of acceptability, appropriateness and feasibility were offered to all participants in the endline assessment. Participants rated their responses on five-point Likert scales, with 5 being the highest (and total scores ranging from 0 to 20). Participants rated the MHL program as acceptable (18.2, SD = 2.8), appropriate (18.1, SD = 2.9) and feasible (17.4, SD = 3.1). Boxplots for the acceptability, appropriateness and feasibility ratings are shown in Figure 2. Regarding user satisfaction, we administered a dichotomous (yes/no) exit interview schedule to all participants in the endline assessments. 99% of participants enjoyed the training, 97% felt satisfied after the program, 96% felt the MHL strategies were easy to understand, 88% found the MHL program therapeutic, 94% found the MHL program to easily fit within their weekly schedule and 97% expressed willingness to use the MHL strategies outside the study e.g. personal challenges and teach others.

### Discussion

#### Summary of key findings

This study describes a local adaptation of *the Guide* (an MHL-based intervention) and evaluates its acceptability, appropriateness, feasibility and initial efficacy among adults residing in urban and rural informal settlements in Kenya. To our knowledge, this is the first study that tests the feasibility of implementing the adapted *Guide* among adults in the community using trained lay facilitators from local grassroots organizations. Findings reveal that the adapted MHL program (*Kenyan Guide*) – with 9 modules and delivered within 3 days by trained lay facilitators – is contextually appropriate and acceptable in urban and rural informal settlements. Feasibility data indicated that the *Kenyan Guide* has the potential to improve participants' MHL, reduce symptoms of common mental health disorders and promote general wellbeing and social support in the short-term. While the observed improvements are promising, it is important to acknowledge that the current study utilized an uncontrolled design. In the absence of a formal control or comparison group, we cannot definitively attribute these changes entirely to the intervention. Our findings are exploratory and preliminary in nature, serving as a foundational evidence base to inform the design of future, more rigorous controlled trials.

#### Comparison with findings from other studies

This is, to our knowledge, the fourth study on *the Guide* to be implemented in the SSA region. The other three studies have been conducted among 153 students aged 15–19 years in Ethiopia (Hassen et al., 2022), 38 school teachers and 4,600 students in Tanzania (Kutcher et al., 2016; Kutcher et al., 2017) and 218 Malawian teachers (Kutcher et al., 2015). Unlike the previous studies which targeted educators and students, our study included adults in the general community (64% women), thus filling an important research gap not only in Kenya but across the SSA region (Atilola, 2015). Most of the extant research on MHL intervention in SSA has targeted school going children and youth, while neglecting those

**Table 3.** Parameter estimates and 95% confidence interval from the random intercept model

*Cambridge Prisms: Global Mental Health*

| Level | Variables | Mental health knowledge score Estimate (95% CI) | p-value | mental health attitude score Estimate (95% CI) | p-value | Depressive symptom score Estimate (95% CI) | p-value | Anxiety symptom score Estimate (95% CI) | p-value | Social support score Estimate (95% CI) | p-value | Wellbeing index score Estimate (95% CI) | p-value |
|---|---|---|---|---|---|---|---|---|---|---|---|---|---|
| **Fixed effects** | | | | | | | | | | | | | |
| Individual | **Sociodemographic characteristics** | | | | | | | | | | | | |
| | **Geographical site** | | | | | | | | | | | | |
| | Kwale | Reference | | Reference | | Reference | | Reference | | Reference | | Reference | |
| | Nairobi | **0.78 (0.41, 1.15)** | **<0.001** | **3.40 (1.34, 5.46)** | **0.001** | **0.89 (0.06, 1.77)** | **0.05** | 0.65 (−0.13, 1.43) | 0.10 | **−3.5 (−5.93, −1.07)** | **0.01** | **2.16 (1.17, 3.2)** | **<0.001** |
| | Mombasa | **0.62 (0.23, 1.01)** | **0.002** | **3.26 (1.09, 5.42)** | **0.003** | **1.71 (0.78, 2.64)** | **<0.001** | **1.44 (0.62, 2.27)** | **0.001** | **−3.96 (−6.52, −1.4)** | **0.002** | 0.7 (−0.37, 1.7) | 0.20 |
| | **Sex** | | | | | | | | | | | | |
| | Female | Reference | | Reference | | Reference | | Reference | | Reference | | Reference | |
| | Male | **0.37 (0.04, 0.69)** | **0.03** | 0.06 (−1.74, 1.86) | 0.95 | **−1.51 (−2.29, −0.74)** | **<0.001** | **−1.01 (−1.7, −0.33)** | **0.004** | 1.19 (−0.93, 3.31) | 0.27 | **1.7 (0.81, 2.55)** | **<0.001** |
| | **Educational level** | | | | | | | | | | | | |
| | None | Reference | | Reference | | Reference | | Reference | | Reference | | Reference | |
| | Primary level | 0.46 (−0.46, 1.38) | 0.33 | 1.80 (−3.29, 6.88) | 0.49 | 1.50 (−0.69, 3.69) | 0.18 | **2.23 (0.20, 4.06)** | **0.03** | −0.35 (−6.35, 5.7) | 0.91 | **−2.7 (−5.1, −0.2)** | **0.03** |
| | Secondary level | **1.35 (0.40, 2.29)** | **0.01** | 4.92 (−0.31, 10.15) | 0.07 | 0.34 (−1.91, 2.60) | 0.77 | 1.23 (−0.76, 3.21) | 0.23 | 2.35 (−3.8, 8.53) | 0.46 | −1.7 (−4.2, 0.83) | 0.19 |
| | Tertiary level | **2.18 (1.19, 3.16)** | **<0.001** | **8.01 (2.56, 13.47)** | **0.004** | −0.35 (−2.70, 2.0) | 0.77 | 0.84 (−1.24, 2.91) | 0.43 | 2.49 (−3.9, 8.93) | 0.45 | −1.3 (−4.0, 1.33) | 0.33 |
| | Age | 0.01 (−0.02, 0.02) | 0.09 | **0.09 (0.01, 0.16)** | **0.02** | −0.01 (−0.04, 0.02) | 0.63 | −0.01 (−0.03, 0.02) | 0.64 | −0.02 (−0.1, 0.7) | 0.72 | −0.01, −0.04, 0.03) | 0.97 |
| | Socio-economic status | **0.13 (0.06, 0.21)** | **0.001** | 0.40 (−0.03, 0.82) | 0.07 | **−0.36 (−0.54, −0.18)** | **<0.001** | **−0.23 (−0.39, −0.1)** | **0.01** | 1.43 (0.93, 1.93) | <0.001 | **0.51 (0.30, 0.72)** | **<0.001** |
| | **Intervention effect** | | | | | | | | | | | | |
| | Pre-intervention | Reference | | Reference | | Reference | | Reference | | Reference | | Reference | |
| | Post-intervention | **1.68 (1.45, 1.92)** | **<0.001** | **5.18 (4.12, 6.23)** | **<0.001** | **−1.22 (−1.06, −0.84)** | **<0.001** | **−0.71 (−1.04, −0.4)** | **<0.001** | **3.33 (2.25, 4.41)** | **<0.001** | **1.132 (0.81, 1.83)** | **<0.001** |
| | Intercept | **8.27 (7.15, 9.38)** | **<0.001** | **21.09 (14.95, 27.23)** | **<0.001** | **9.37 (6.73, 12.01)** | **<0.001** | **6.49 (4.15, 8.82)** | **<0.001** | **54.18 (46.94, 61.42)** | **<0.001** | **11.80 (8.83, 14.78)** | **<0.001** |
| **Random effects** | | | | | | | | | | | | | |
| Individual | Residual variance (σ²ε) | 4.47 | – | 88.23 | – | 11.53 | – | 8.50 | – | 91.56 | – | 20.58 | – |
| Group | Random intercept variance (σ²μ) | 1.29 | – | 63.41 | – | 14.17 | – | 11.28 | – | 104.08 | – | 14.96 | – |

Bold p-values are those which were significant at the ≤0.05 level.

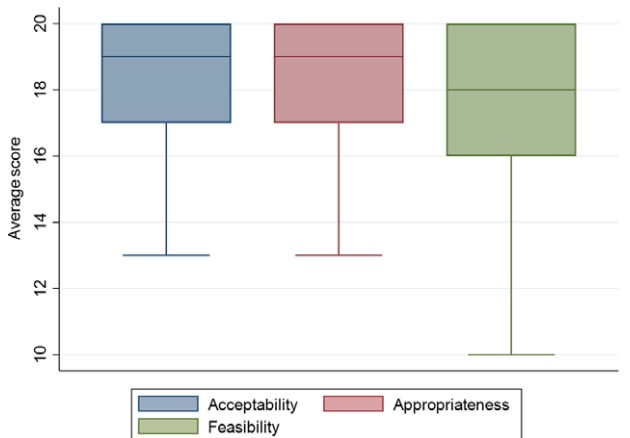

**Figure 2.** Box plots of average acceptability, appropriateness and feasibility ratings of the MHL program.

residing in the general population despite low levels of MHL and high burden and prevalence of mental health problems in these individuals, especially among women. Despite a difference in the population targeted, our findings in the current study are largely consistent with those of the previous studies. Our results are also comparable to previous findings in Kenya. Mutiso and colleagues utilized the World Health Organization mhGAP-Intervention Guide as an educational tool for one-on-one contact in a clinical setting to increase literacy on the specified mental disorders and concluded that the guide was a feasible tool to improve MHL in low-resource settings. Similar findings have been reported elsewhere in Ghana among community leaders (Arthur et al., 2022).

In resource-constrained settings such as those in LMICs, the use of trained and supervised lay facilitators for the delivery of mental health/psychological interventions has been preferred due to the acute shortage of mental health specialists (Van Ginneken et al., 2011). The high acceptability and successful implementation of the *Kenyan Guide* by lay facilitators from local grassroots organizations suggests that lay facilitators can adequately deliver MHL programs after training and continued supervision. Additionally, our partnership with local community-based organizations who were sub-awarded to implement the MHL program is a crucial strategy for several reasons: a) ensures trust and cultural sensitivity, b) improved access and engagement, c) fosters a sense of ownership and responsibility, leading to more sustainable solutions, d) benefits from community insights and e) enhances peer support and empowerment. For instance, the grassroots organizations involved noted that they were able to strengthen their organizational capacity in mental health programming. Through the MHL training, mental health tools and collaborative networks, the local organizations enhanced their mental health advocacy efforts in the community, improved their mental health referral pathways, enhanced peer-to-peer learning and increased their visibility and credibility with other partners, doners, community members/stakeholders, which have opened doors for new collaborations, technical support and funding. Indeed, evidence suggests that community-engaged mental health and wellbeing initiatives in resource-limited settings have the potential to enhance mental health outcomes and wellbeing when they actively involve community members (Chutiyami et al., 2025).

The high recruitment of participants into the study also supports the feasibility of enrolling community participants into the MHL program. Furthermore, the Implementation of the *Kenyan*

*Guide* within common community spaces e.g. social halls, public schools and religious institutions potentially reduces the implementation cost e.g. by avoiding costly venue charges and ensures that the community members receive the program within their areas of work/residences thus reducing costs of travel.

In this study, discussions with different stakeholders and the comprehensive formative phase research findings (Mwangala et al., 2025) ensured that the *Kenyan Guide* was contextually appropriate. When implemented, we observed significant reductions in symptoms of CMDs and improvement in general wellbeing and social support at endline assessment. This proposition is supported by the positive experiences reported by the respondents in this study. These are also important preliminary findings which can inform future MHL work in Kenya and similar contexts.

### Interpretation of the current findings and implications for future work

From the findings of this project, the adapted MHL program, *the Kenyan Guide* appears contextually appropriate within the urban and informal settlements in Kenya. The partnership with local grassroots organizations to implement the MHL program is highly acceptable, and so is the use of lay facilitators. In these settings, it is possible to recruit and retain community members in the adapted MHL program, as an excellent response and retention rates were observed. It is also feasible to deliver the MHL program over 3 days. Preliminary feasibility data shows that *the Kenyan Guide* has the potential to enhance MHL, reduce symptoms of CMDs and improve general wellbeing and social support of the community members. Our findings in this study justify the need for a fully powered, definitive randomized controlled trial in the future, to fully assess the efficacy of the adapted MHL program in Kenya or similar settings. Future work should also examine the factors that are likely to influence the implementation and sustainability of this program in different settings. Successful MHL programs are likely to increase the number of people who seek mental health services in these areas. While the MHL landscape is evolving, significant work remains to be done to ensure that mental health systems are adequately prepared for the increasing MHL, especially in LMICs. This includes strengthening the mental health referral pathway, addressing the workforce shortages, increasing access to care e.g. through digital mental health, integrating mental health into primary care, and community engagement.

### Limitations

The current study findings should be interpreted considering a few limitations. The pre/post design carries with it inherent challenges compared to other experimental designs. It is not possible to fully attribute the improvements in MHL, mental health and wellbeing outcomes over the course of this intervention to the intervention itself. Nonetheless, the timeline of the assessments – immediately before and after the intervention – make other potential explanations for the observed improvements quite unlikely. However, the short-term follow up limits our understanding of the long-term impact of the *Kenyan Guide*. The study design cannot also evaluate the improvements in the lay facilitators' MHL retention over time. Follow-up studies can ascertain to what extent the lay facilitators are able to retain the skills and their application. The oversampling of women participants into study may have introduced selection bias. Future studies should include large enough samples for each

sex to broadly generalize the findings across sex. Additionally, we cannot rule out potential self-report bias, as all data were collected using self-report measures. However, all measures were adequately adapted and/or validated before use in the study setting, and all the facilitators were thoroughly trained. Future studies may consider using independent assessors blinded to the study to minimize potential social desirability and interviewer biases.

## Conclusions

Overall, the adapted MHL program was found to be acceptable and appropriate in the study setting and could feasibly be delivered by trained non-specialist facilitators from local grassroots organizations, making it a promising approach to enhance the MHL of adults in the community in this setting. We believe that our study serves as an important benchmark for community MHL programs in similar contexts given its comprehensive cultural adaptation, rigorous implementation, ethical and participatory approaches. Preliminary findings show that the adapted MHL program has the potential to increase participants' MHL and improve mental health and wellbeing outcomes in the short term. Future research should utilize controlled trials to corroborate our findings and explore how improvements can be sustained over the long term, e.g. by including assessment points at 3 to 6 months. Such longitudinal data is important to determine whether the observed changes can be maintained over time or if booster sessions are required to prevent a return to baseline levels.

**Open peer review.** To view the open peer review materials for this article, please visit http://doi.org/10.1017/gmh.2026.10140.

**Supplementary material.** The supplementary material for this article can be found at http://doi.org/10.1017/gmh.2026.10140.

**Data availability statement.** The de-identified dataset used and analyzed during this current study will be made available upon reasonable request, in due consideration of Aga Khan University's data sharing policies. Requests to access the datasets should be directed to the Research Office Aga Khan University Kenya through research.supportea@aku.edu or akukenya.researchoffice@aku.edu.

**Acknowledgements.** The authors would like to thank all the stakeholders and participants who contributed to this study. Special appreciation goes to the implementing CSOs including Tuwajali Wajane Kwale Initiative, Stretchers Youth Organization, Kwale Focus Empowerment, Kamili Organization, Girls for Girls Africa Mental Health Foundation and Otto Benecker Foundation, and the county government representatives from Nairobi, Mombasa and Kwale counties for their invaluable support in providing local expertise.

**Author contribution.** AA conceptualized and obtained funding for the study. AA and PNM developed study protocols, tools and training programs. VRK led the training of the representatives of local CSOs on the adapted MHL program. PNM, GM, NS and AK coordinated the piloting of the adapted MHL program and the data collection process. PNM wrote the first draft of the manuscript. All authors critically reviewed subsequent versions of the manuscript and approved the final version for submission. All authors read and approved the final manuscript.

**Financial support.** This publication was produced with the financial support of Global Affairs Canada (grant number P_007597). AA and PNM are also supported by the Science for Africa Foundation to [Ref: Del-22-002] with support from Wellcome Trust and the UK Foreign, Commonwealth & Development Office and is part of the EDCTP 2 program supported by the European Union. For purposes of open access, the author has applied a CC BY public copyright license to any Author Accepted Manuscript version arising from this submission. The funders had no role in the study's design, in the collection, analyses or interpretation of data, in the writing of the manuscript or in the decision to publish the results.

**Competing interests.** The authors declare none.

**Ethics statement.** The project was carried out in accordance with ethical principles and guidelines for involving human participants as stipulated in the Helsinki Declaration. Ethical approval was sought and given by the Aga Khan, Nairobi Institutional Scientific and Ethics Review Committee (Ref: 2023/ISERC-37(v2)). Permission to conduct the study in Kenya was granted by the National Commission for Science, Technology and Innovation (Ref: NACOSTI/P/23/28548). Respective departments of health across the 3 counties also gave local permits to carry out the study in the respective counties: Mombasa (Ref: MCG/COPH/RCH. /111), Nairobi (Ref: NCCG/DHS/REC/240) and Kwale (Ref: CG/KWL/6/6/1/CECM/39/VOL.1/34). All the participants provided written informed consent for their participation in the study.

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
