## [Reviewer Report]

Thank you for the opportunity to review the paper that contributes to what we currently know about MHL programs for Kenya and LAMICs. The paper is well written and findings are well articulated. I have made a few suggestions on methods for you to consider as per below:

Impact Statement

The opening sentence (lines 5-9) makes claims that are well beyond the typical scope of a MHL program. Please refer to the definition of MHL and revise this statement.

Introduction

Excellent background that links MHL concepts and appraisal of recent literature.

Methods

Authors state that participants were required to be at least 18 years old (lines 19-22). Considering preventive/promotive rationale provided in the introduction section, and increasing evidence of rising mental illness burden among child and adolescent categories, what was the justification for excluding them? And noting robust and ethical research must include processes to mediate consent and language issues.

Authors state that researchers intentionally sampled more women because the study was ‘part of a larger project aimed to advance gender equality and address challenges faced by women’ (lines 19-25). Reviewer finds this confounding, and potentially detracts on the claims the authors can make on their research as being unbiased exploration of ‘feasibility, acceptability and initial efficacy of a community-based MHL’…

• Is the title not potentially misleading

• Was the research unbiased

• Why is this (potential bias) not acknowledged in limitations section of the paper?

---

## [Reviewer Report]

Dear authors,

Thank you for the opportunity to read your article.

This is a well-conceived and clearly written study.

The topic is relevant to global mental health research and to public health and makes a valuable contribution to the emerging field of implementation science in mental health in sub-Saharan Africa.

The conclusions are appropriately cautious and consistent with the data presented.

Although the study present some strengths, namely its relevance and novelty, its cultural adaptation and detailed implementation, its rigorous measurement and its ethical and participatory approach, there are some improvements that could make it a benchmark for community mental health literacy programmes in similar contexts.

Below are a few suggestions that may contribute to improving the study:

- Consider shortening the introduction slightly — the literature review is strong but can be more concise;

- Control group - without a control or comparison group, it is not possible to attribute observed changes solely to the intervention I recommend that you acknowledge more explicitly in the discussion and emphasize the exploratory nature of findings. Future trials should include randomization or matched controls;

- Follow-up assessment - all outcomes were measured immediately post-intervention; you can suggest in your study the inclusion of at least one follow-up point (e.g., 3–6 months) to test sustainability of effects;

- Sample size - even for pilot studies, a rationale (based on feasibility or effect size expectations) should be provided (lines 19 to 21 of page 21);

- Analytical depth - only paired t-tests were used; no adjustment for covariates or potential confounders; I recommend that you consider multivariate regression or mixed models to adjust for gender, education, and county effects;

- Potential self-report bias - all data are self-reported, collected by facilitators involved in the intervention; it will be interesting that you discuss possible social desirability and interviewer bias; independent assessors could be used in future studies.

In my opinion the paper convincingly shows that the adapted “Kenyan Guide” is acceptable, appropriate, and feasible. The short-term outcomes justify further investigation through a fully powered randomized controlled trial.

---

## [Reviewer Report]

- The authors must please ensure that for terms that have acronyms such as mental health literacy (MHL) and SSA (sub-Saharan Africa) once they provided the terms in full and provided the acronym in brackets, they should then use the acronym throughout. It was noted that there was a mix, as authors used the acronym in some cases and the full keywords in other cases. Please ensure consistency.

- For the definition of MHL by Jorm et al (1997) the authors must also provide the page number of the source as they placed the definition in brackets to show that they took the words directly from the authors.

- The authors indicated that both English and Swahili were used in the study. The question is, the translation of the work from English to Swahili and the back-translation from Swahili to English were done by who? this must be clarified as an independent Swahili language expert was supposed to be used to avoid incorrect substitution of words and also for credibility purposes.

- The authors must also ensure that line spacing of all paragraphs is consistent throughout the manuscript. Some inconsistencies were noted.

- The discussion section can be more nuanced so that it has a narrative flow and fluidity while it is linked to existing literature so that the reader can be taken into confidence regarding how the results fare within the context of existing literature.

- The authors must also ensure that at the reference list DOI’s are provided as hyperlinks for journal articles and links for other sources so that such sources can be easily verifiable through these hyperlinks.

- Some minor sentence construction and grammar issues were noted - the authors must proof-read the manuscript for correctness.

- Despite the minor issues that the authors have to attend to, the authors investigated an important area especially within the sub-Saharan Africa and the noviceness of their study is clearly evident.

---

## [Editor Report]

Dear Dr Mwangala

Thank you for your submission to Cambridge Prisms: Global Mental Health. Following a period of review, three independent reviewers have communicating their decisions and we can now confirm that the decision of Major Revision has been reached. Kindly address each comment made by each reviewer, and submit a revised manuscript. We are looking forward receiving you revision.

Best regards,

---

## [Reviewer Report]

It is my opinion that the changes introduced meet and respond to the reviewers' comments.

Congratulations on your work.